# MEDIAN DC FOR SIGN RECOVERY: PRIVACY CAN BE ACHIEVED BY DETERMINISTIC ALGORITHMS

## ABSTRACT

Privacy-preserving data analysis becomes prevailing in recent years. It is a common sense in privacy literature that strict differential privacy can only be obtained by imposing additional randomness in the algorithm. In this paper, we study the problem of private sign recovery for sparse mean estimation and sparse linear regression in a distributed setup. By taking a coordinate-wise median among the reported local sign-vectors, which can be referred to as a median divide-and-conquer (Med-DC) approach, we can recover the signs of the true parameter with a provable consistency guarantee. Moreover, without adding any extra randomness to the algorithm, our Med-DC method can protect data privacy with high probability. Simulation studies are conducted to demonstrate the effectiveness of our proposed method.

## 1 INTRODUCTION

With the development of technology for data acquisition and storage, the modern dataset has a larger scale, more complex structure, and more practical considerations, which addresses new challenges for data analysis. In recent years, large quantities of sensitive data are collected by individuals and companies. While one wants to extract more accurate statistical information from the distributed dataset, we must also beware of the leakage of this sensitive personal information during the training process. This calls for the study of distributed learning under privacy constraints (Pathak et al., 2010; Hamm et al., 2016; Jayaraman et al., 2018).

In privacy literature, differential privacy, which was firstly proposed in Dwork et al. (2006), has been the most widely adopted definition of privacy tailored to statistical data analysis. It has achieved tremendous success in real-world applications. Denote the data universe to be $\mathcal{X}$, for the dataset $\mathbb{X} = \{\boldsymbol{X}_i\}_{i=1}^n \in \mathcal{X}^n$ where $\boldsymbol{X}_i$'s are the data observations. The $(\epsilon, \delta)$-differentially privacy can be defined as follows.

**Definition 1.** *(Differential Privacy in Dwork et al. (2006)) A randomized algorithm $\mathcal{A} : \mathcal{X}^n \to \Theta$ gives $(\epsilon, \delta)$-differentially private if for any pair of adjacent datasets $\mathbb{X} \in \mathcal{X}^n$ and $\mathbb{X}' \in \mathcal{X}^n$, there always holds*

$$\mathbb{P}\big\{\mathcal{A}(\mathbb{X}_{1:n}) \in U\big\} \leq e^\epsilon \cdot \mathbb{P}\big\{\mathcal{A}(\mathbb{X}'_{1:n}) \in U\big\} + \delta, \tag{1}$$

*for every subset $U \subseteq \Theta$.*

Here two datasets $\mathbb{X}$ and $\mathbb{X}'$ are adjacent if and only if the Hamming distance (Lei, 2011) of these two datasets of same size $H(\mathbb{X}, \mathbb{X}') = 1$. As we can see, the quantities $\epsilon$ and $\delta$ measure the level of privacy loss. There are also several relaxations of differential privacy (see, e.g., Bun & Steinke (2016); Dwork & Rothblum (2016); Mironov (2017); Dong et al. (2019)) designed for the ease of analysis. However, in these definitions, the dataset $\mathbb{X}$ is always assumed to be fixed, and the probability in (1) only takes over the randomness of the algorithm $\mathcal{A}$. Therefore, it is impossible to achieve strict differential privacy without adding auxiliary perturbations in the algorithm. On the other hand, the statistical performance of the output is inevitably deteriorated by the additional randomness. This lead to a large body of works discussing the tradeoff between accuracy and privacy (Wasserman & Zhou, 2010; Bassily et al., 2014; Bun et al., 2018; Duchi et al., 2018; Cai et al., 2019).

In this paper, we consider the private sign recovery problem in the distributed system. To be more precise, assume the parameter of interest is a sparse vector, which has many zeros in its entries.

The task is to identify the signs of the parameter from the observations stored in multiple machines while protecting each individual's privacy. The sign recovery problem, as an extension of sparsity pattern recovery, has found its significance in a broad variety of contexts, including variable selection (Tibshirani, 1996; Miller, 2002), graphical models (Meinshausen & Bühlmann, 2006; Cai et al., 2011), compressed sensing (Candes & Tao, 2005; Donoho, 2006), and signal denoising (Chen et al., 2001). However, this problem is rarely considered in the privacy community.

To address the sign recovery problem, we propose the Median Divide-and-Conquer (Med-DC) method, a simple two-step procedure. Firstly, each local machine estimates the sparse parameter and sends the sign-vectors back to the server; Secondly, the server aggregates these sign-vectors using coordinate-wise median and output the final sign estimator. While mean based divide-and-conquer (also referred to as Mean-DC) approaches have been widely analyzed in distributed learning literature (Mcdonald et al., 2009; Zhang et al., 2013; Lee et al., 2017; Battey et al., 2018), the median-based counterpart has not yet been well explored. It is well-known that naively averaging the local estimators behaves badly for nonlinear and penalized optimization problems. This is because averaging cannot reduce the bias in the local sub-problems. In particular, for the distributed Lasso problem, as mentioned in Lee et al. (2017), the estimation error of averaged local Lasso estimator is of the same order as that of local estimators. However, when only considering the sign recovery problem, we found that the Med-DC method perfectly fits the nature of the distributed private setup. (See Section 2.2 for more detailed discussions) For the sake of clarity, we only consider the sign recovery problem for sparse mean estimation and sparse linear regression, the two fundamental models in statistics.

The proposed Med-DC method has the following advantages:

• **Consistent recovery.** For both sparse mean estimation and sparse linear regression, the Med-DC method consistently recovers the signs of the true parameter with theoretical guarantees. Under some constraints, we can prove that our approach can identify signals larger than $C\sqrt{\log n/N}$ for some constant $C > 0$ (where $N$ is the full sample size and $n$ is the local sample size), which coincides with the minimal signal level in the single machine setting (all data stored in one machine).

• **Efficient communication.** To recover the signs of the parameter of interest in the distributed setup, a naive approach is to estimate the parameter using existing private distributed estimation methods and take the signs of the estimators. However, these methods usually involve multi-round aggregation of gradient information or local estimators, which seems costly for the simple sign recovery problem. Instead, our approach only aggregates the vectors of signs (bits information) in one shot, which is much more communicationally efficient.

• **Weak privacy.** By relaxing the differential privacy to high-probability sense, our deterministic Med-DC method can be proved to be weakly 'private'. We also extend this concept to group privacy. To the best of our knowledge, this is the first deterministic algorithm that has a provable high-probability privacy guarantee. Moreover, as each machine only needs to transmit the vectors of signs, instead of the local estimators or gradient vectors, our proposed method also protects the privacy of each local machine, since gradient sharing can also result in privacy leakage (Zhu et al., 2019).

• **Wide applicability.** We believe the Med-DC approach deserves more attention due to its excellent practical performance and ease of implementation. For example, it is promising to apply the Med-DC method to wider classes of models, (*e.g.* , generalized linear model, $M$-estimation, etc.) or hybridize this method with many sophisticated distributed algorithms like averaged de-biased estimator in Lee et al. (2017) and Communication-Efficient Accurate Statistical Estimator (CEASE) in Fan et al. (2019).

**Notations.** For every vector $\boldsymbol{v} = (v_1, ..., v_p)^T$, denote $|\boldsymbol{v}|_2 = \sqrt{\sum_{l=1}^{p} v_l^2}$, $|\boldsymbol{v}|_1 = \sum_{l=1}^{p} |v_l|$, and $|\boldsymbol{v}|_\infty = \max_{1 \le l \le p} |v_l|$. Moreover, we use $\mathrm{supp}(\boldsymbol{v}) = \{1 \le l \le p \mid v_l \ne 0\}$ as the support of the vector $\boldsymbol{v}$, and $\boldsymbol{v}_{-l} = (v_1, \ldots, v_{l-1}, v_{l+1}, \ldots, v_p)^T$. For every matrix $\boldsymbol{A} \in \mathbb{R}^{p_1 \times p_2}$, define $\|\boldsymbol{A}\| = \sup_{|\boldsymbol{v}|_2=1} |\boldsymbol{A}\boldsymbol{v}|_2$, $\|\boldsymbol{A}\|_\infty = \max_{1 \le l_1 \le p_1, 1 \le l_2 \le p_2} |A_{l_1, l_2}|$, $\|\boldsymbol{A}\|_{L_\infty} = \sup_{|\boldsymbol{v}|_\infty=1} |\boldsymbol{A}\boldsymbol{v}|_\infty$ as various matrix norms, $\Lambda_{\max}(\boldsymbol{A})$ and $\Lambda_{\min}(\boldsymbol{A})$ as the largest and smallest eigenvalues of $\boldsymbol{A}$ respectively. We will use $\mathbb{I}(\cdot)$ as the indicator function and $\mathrm{sgn}(\cdot)$ as the sign function. For two sequences $a_n, b_n$, we say $a_n \asymp b_n$ when $a_n = O(b_n)$ and $b_n = O(a_n)$ hold at the same time. For simplicity, we denote $\mathbb{S}^{p-1}$ and $\mathbb{B}^p$ as the unit sphere and unit ball in $\mathbb{R}^p$ centered at $\boldsymbol{0}$. For a sequence of vectors $\{\boldsymbol{v}_i\}_{i=1}^{n} \subseteq \mathbb{R}^p$, we denote $\mathrm{med}(\cdot)$ as the coordinate-wise median. Lastly, the generic constants are assumed to be independent of $m, n$, and $p$.

## 2 PRIVATE SIGN RECOVERY OF MEAN VECTOR

### 2.1 MEDIAN BASED DIVIDE-AND-CONQUER

Let $\boldsymbol{\mu}^* = (\mu_1^*, \ldots, \mu_p^*)^{\mathrm{T}}$ be the true parameter of interest. We assume the vector is sparse in the sense that many entries $\mu_l^*$ are zero. There are $N$ i.i.d. observations $\boldsymbol{X}_i$'s satisfying $\mathbb{E}[\boldsymbol{X}_i] = \boldsymbol{\mu}^*$, and they are evenly stored in $m$ different machines $\mathcal{H}_j$ (where $1 \leq j \leq m$). Denote $\mathbb{X} = \{\boldsymbol{X}_1, \ldots, \boldsymbol{X}_N\}$ as the full dataset. For simplicity we assume $N = mn$ so that each machine has equally $n$ samples. Our task is to identify the signs of $\boldsymbol{\mu}^*$ on all coordinates (denoted as $\mathrm{sgn}(\boldsymbol{\mu}^*)$) in this distributed setup while protecting the privacy of every element $\boldsymbol{X}_i$ on each machine $\mathcal{H}_j$.

To recover the signs of the true mean vector $\boldsymbol{\mu}^*$ privately, the most direct way is to estimate the mean by some existing differentially private algorithms and take the signs of the estimator coordinate-wisely. By post-processing property of differential privacy (Proposition 2.1 in Dwork & Roth (2014)), we know this sign recovery method is also differentially private. Private mean estimation is a fundamental problem in private statistical analysis and has been studied intensively (Dwork et al., 2006; Lei, 2011; Bassily et al., 2014; Cai et al., 2019). The standard approach is to project the data onto a known bounded domain, and then apply the noises according to the diameter of the feasible domain and the privacy level. However, this method requires input data or the true parameter lies in a known bounded domain, which seems unsatisfactory in practice. Moreover, since we only want to estimate the signs, which take value in the discrete set $\{-1, 0, 1\}$, it seems unnecessary to perturb the mean directly.

Indeed, to recover the signs of the true parameter, there is no need to obtain an accurate mean with all data. Instead, we propose a Median based Divide-and-Conquer (Med-DC) approach. To be more precise, we can estimate the signs on each local machine $\mathcal{H}_j$, and aggregate these vectors of signs by taking median to produce a more accurate sign estimator.

To present our method more clearly, we define the following quantization function $\mathcal{Q}_\lambda(\cdot)$,

$$\mathcal{Q}_\lambda(x) = \mathrm{sgn}\Big\{\underbrace{\mathrm{sgn}(x) \cdot (|x| - \lambda)_+}_{\text{shrinkage operator}}\Big\} = \begin{cases} \mathrm{sgn}(x) & \text{if } |x| > \lambda, \\ 0 & \text{if } |x| \leq \lambda. \end{cases} \tag{2}$$

Here $\lambda$ is a thresholding parameter. When $x$ is a vector, $\mathcal{Q}_\lambda(x)$ performs the above operation coordinate-wisely. In particular, when $\lambda = 0$, the function $\mathcal{Q}_0(\cdot)$ acts the same as the sign function $\mathrm{sgn}(\cdot)$. Then we present our method in Algorithm 1.

---

**Algorithm 1** Median divide-and-conquer (Med-DC) for sparse mean estimator.

---

**Input:** Dataset $\mathbb{X} = \{\boldsymbol{X}_1, \ldots, \boldsymbol{X}_N\}$ evenly divided into $m$ local machines $\mathcal{H}_j$ (where $j = 1, \ldots, m$), the universal thresholding parameter $\lambda_N$.

1: **for** $j = 1, \ldots, m$ **do**
2:     The $j$-th machine $\mathcal{H}_j$ computes the local sample mean $\bar{\boldsymbol{X}}_j = n^{-1} \sum_{i \in \mathcal{H}_j} \boldsymbol{X}_i$. Then $\mathcal{H}_j$     sends $\boldsymbol{Q}_j = \mathcal{Q}_{\lambda_N}(\bar{\boldsymbol{X}}_j)$ to the server.
3: **end for**
4: The server takes coordinate-wise median

$$\hat{\boldsymbol{Q}}(\mathbb{X}) = \mathrm{med}(\mathcal{Q}_{\lambda_N}(\bar{\boldsymbol{X}}_j) \mid 1 \leq j \leq m), \tag{3}$$

**Output:** The vector of signs $\hat{\boldsymbol{Q}}(\mathbb{X})$.

---

The choice of thresholding parameter will be discussed after Theorem 1 in Section 2.3. Especially mention that there are some cases when the median is not uniquely determined. For example, it is possible that there are the same numbers of 0's and 1's, then the median can be arbitrary value in $[0, 1]$. To avoid ambiguity, we simply take $\hat{Q}_l(\mathbb{X}) = 0$ (where $l$ denotes the coordinate index) whenever the median $\hat{Q}_l(\mathbb{X})$ is not unique.

Another important remark is that, the proposed sign recovery algorithm $\hat{\boldsymbol{Q}}(\cdot)$ is deterministic. More precisely, as there is no additional random perturbation in Algorithm 1, the output $\hat{\boldsymbol{Q}}(\mathbb{X})$ is completely determined by the input dataset $\mathbb{X}$. It is able to protect data privacy in a weaker sense.

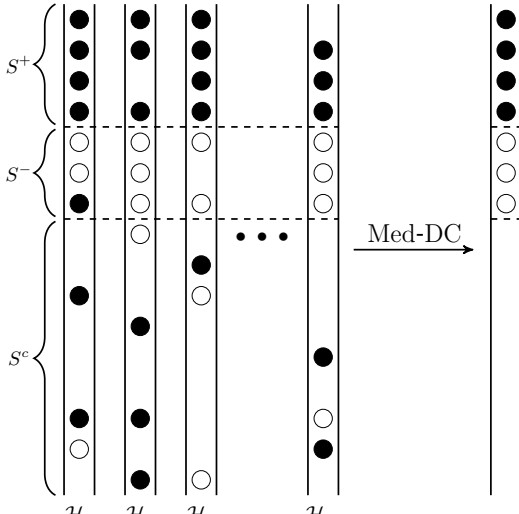

Figure 1: This figure visualizes the mechanism of the median divide-and-conquer (Med-DC) approach. Denote $S^+$, $S^-$ and $S^c$ as the sets of positives, negatives, and zeros of the true parameter $\boldsymbol{\mu}^*$ respectively. The black dots and white dots on each column represent the estimated positive and negative locations on each local machine.

## 2.2 Intuition Behind Med-DC

Before presenting the theoretical results of our Med-DC method, we briefly illustrate the intuition behind it. The median mechanism among the collection of discrete values in $\{-1, 0, 1\}$ can be equivalently regarded as a voting game. At each coordinate, we take the element which gets more than half of the votes, and we take it as $0$ when there is no such element. The mechanism of the Med-DC method can be visualized in Figure 1.

**Recovery Consistency.** The key insight of the Med-DC method is that, according to Berry-Essen theorem, the distribution of sample means on local machines is close to normal distribution centered at the true parameter $\boldsymbol{\mu}^*$. Therefore, on each coordinate $l$, the local sample means $\{\bar{X}_{1,l}, \ldots, \bar{X}_{m,l}\}$ are approximately symmetrically distributed around $\mu_l^*$. Based on this observation, the sign recovery consistency of the median mechanism becomes clear: For $\mu_l^* > \lambda_N$ ($\mu_l^* < -\lambda_N$), it is likely to have at least $m/2$ elements larger than $\lambda_N$ (smaller than $-\lambda_N$), which makes the median of local signs more inclined to be $1$ ($-1$). For $\mu_l^* = 0$, by the approximate symmetry of local sample means, the numbers of $1$'s and $-1$'s are likely to be equal. Thus the median tends to be $0$.

**Weak Privacy.** Given the dataset $\mathbb{X}$ and the adjacent datasets $\mathbb{X}'$, when applying Algorithm 1 to $\mathbb{X}'$, there would be at most one element among the set of signs $\{\mathcal{Q}_{\lambda_N}(\bar{\boldsymbol{X}}_j)\}_{j=1}^m$ change. With high probability, the change of one element would not affect the median value of $\hat{\boldsymbol{Q}}(\mathbb{X})$. This interpretation is conceptually coincident with the idea of differential privacy in Definition 1. However, this 'privacy' guarantee does not hold for all data. For instance, in one-dimensional case, it is possible that $\mathbb{X}$ produces $(m/2)$ positives and $m/2$ negatives ($\hat{\boldsymbol{Q}}(\mathbb{X}) = 0$), and the adjacent dataset $\mathbb{X}'$ produces $m/2 + 1$ positives and $m/2 - 1$ negatives ($\hat{\boldsymbol{Q}}(\mathbb{X}') = 1$), which contradict with standard definition of differential privacy. However, by Proposition 1, we can show that the Med-DC method guarantees privacy in a weaker sense with probability tending to $1$, which implies that the aforementioned unpleasant case only appears with a small probability.

**Connection with Robust Statistics.** Our Med-DC method has an intimate connection with the median-of-means (MOM) estimator, a robust mean estimator which has attracted considerable recent interests in statistics and machine learning communities (Nemirovsky & Yudin, 1983; Yin et al., 2018; Minsker, 2019; Lecué & Lerasle, 2020). To estimate the mean, the MOM estimator takes the median among the local sample means. Both the MOM estimator and our Med-DC method use the

symmetrization effect of the local average. As our task is to find the signs of $\boldsymbol{\mu}^*$, we only aggregate the signs of each local estimator.

Robust statistics studies the estimator that is not much influenced by a small portion of data, which is conceptually similar to differential privacy. Indeed, the connection between robustness and privacy has been pointed out in Dwork & Lei (2009); Smith (2011); Avella-Medina (2019); Brunel & Avella-Medina (2020). In particular, Brunel & Avella-Medina (2020) leverages the MOM estimator and "Propose-Test-Release" (PTR) framework in Dwork & Lei (2009) to develop private mean estimator without any boundedness assumptions on the data and parameter. It is worthwhile noting that, we can also combine the PTR-framework with our Med-DC approach to develop a strictly differentially private sign estimator.

## 2.3 THEORY OF MEAN VECTOR SIGN RECOVERY

To discuss the theoretical properties of our method, we introduce the distribution space $\mathcal{P}$ of $\boldsymbol{X}$

$$\mathcal{P}(\boldsymbol{\mu}^*, C) = \left\{ \mathbb{P} \Big| \mathbb{E}_{\mathbb{P}}[\boldsymbol{X}] = \boldsymbol{\mu}^*, \max_{1 \le l \le p} \mathbb{E}_{\mathbb{P}}\big[|X_l - \mu_l^*|^3\big] \le C \right\}, \tag{4}$$

where $C > 0$ is some constant. This is a rather weak condition on the distribution of $\boldsymbol{X}$. At each coordinate, we assume $X_l$ has a finite third-order moment, which is common in median-of-mean literature (Minsker, 2019). Then we have the sign consistency of the proposed estimator $\hat{\boldsymbol{Q}}(\mathbb{X})$.

**Theorem 1.** *(Sign consistency of Med-DC) Let $N = mn$ i.i.d. random vectors $\{\boldsymbol{X}_1, \ldots, \boldsymbol{X}_N\}$ sampled from $\mathcal{P}(\boldsymbol{\mu}^*, C)$ be evenly distributed in $m$ subsets $\mathcal{H}_1, \ldots, \mathcal{H}_m$. Moreover, there are sufficiently large constants $C_1, C_2, \gamma_0 > 0$, such that*
   *(a) The dimension $p$ satisfies $p = O(n^{\gamma_0})$, and take $\lambda_N = C_1(\sqrt{\log n/N} + 1/n)$;*
   *(b) Denote $S = \operatorname{supp}(\boldsymbol{\mu}^*)$, then there is $\min_{l \in S} |\mu_l^*| \ge C_2 \lambda_N$.*
*Then $\hat{\boldsymbol{Q}}(\mathbb{X})$ defined in Algorithm 1 satisfies that, for some large $\gamma_1$ depends on $C_1, C_2, \gamma_0$, there is*

$$\mathbb{P}\Big(\hat{\boldsymbol{Q}}(\mathbb{X}) = \operatorname{sgn}(\boldsymbol{\mu}^*)\Big) \ge 1 - n^{-\gamma_1}.$$

As we can see from assumptions (a) and (b), when $m = O(n)$, the thresholding parameter $\lambda_N$ can be chosen to be $C_1\sqrt{\log n/N}$, which means our algorithm can identify the signal above the order of $O(\sqrt{\log n/N})$. This is coincident with the optimal signal-to-noise ratio in a single machine setting (all $N$ data are stored in one machine). Moreover, we note that the assumptions in Theorem 1 does not really require the true parameter to be 'sparse' in the sense that $s \ll p$. Instead, we only assume there is a fixed gap (at the level of $O(\sqrt{\log n/N})$) between zeros and those nonzero elements. In the following proposition, we show that our algorithm can protect data privacy with high probability.

**Proposition 1.** *(Privacy of Med-DC) Under the same assumptions as Theorem 1, denote $\mathcal{D}(\mathbb{X})$ as the collection of datasets $\mathbb{X}'$ adjacent to $\mathbb{X}$, then for some large $\gamma_2 > 0$, there is*

$$\mathbb{P}\Big(\hat{\boldsymbol{Q}}(\mathbb{X}) = \hat{\boldsymbol{Q}}(\mathbb{X}'), \text{ for all } \mathbb{X}' \in \mathcal{D}(\mathbb{X})\Big) \ge 1 - n^{-\gamma_2}. \tag{5}$$

It is worth noting that, the privacy guarantee formulated in equation (5) differs with the standard definition of differential privacy (see Definition 1) in several aspects. Firstly, as the algorithm $\hat{\boldsymbol{Q}}(\cdot)$ is deterministic, the randomness in (5) comes from the data generating mechanism. It implies that this algorithm preserves data privacy with probability tending to 1, and precludes some extreme cases, which may happen with a small probability. This is essentially different from the standard definition of differential privacy and its variants, which always assume the dataset is fixed, and the randomness comes from the algorithm itself. Secondly, combining Theorem 1 and Proposition 1, we know that with probability tending to 1, this algorithm recovers the true signs, and the modification of a single entry of $\mathbb{X}$ does not affect the output at all (0 privacy loss). Therefore, this method can be roughly regarded as a $(0,0)$-differentially private algorithm.

## 3 Private Sign Recovery of Linear Regression

### 3.1 Med-DC of Regression Parameter

Let $(\boldsymbol{X}_i, Y_i)$ (where $i = 1, \ldots, N$) be i.i.d. observations from the model

$$Y = \boldsymbol{X}^{\mathrm{T}} \boldsymbol{\theta}^* + z, \tag{6}$$

where $\boldsymbol{\theta}^* = (\theta_1^*, \ldots, \theta_p^*)^{\mathrm{T}}$ is the true sparse regression parameter, and $z$ is the noise independent with the covariate $\boldsymbol{X}$. Denote the full dataset as $\mathbb{X} = \{(\boldsymbol{X}_1, Y_1), \ldots, (\boldsymbol{X}_N, Y_N)\}$, and $\mathbb{X}$ is evenly divided into $m$ local machines $\mathcal{H}_j$ (where $1 \le j \le m$). Similarly, we attempt to recovery the vector of signs $\mathrm{sgn}(\boldsymbol{\theta}^*) = (\mathrm{sgn}(\theta_1^*), \ldots, \mathrm{sgn}(\theta_p^*))^{\mathrm{T}}$.

Sparse linear regression is an important topic in the statistical literature. The least absolute shrinkage and selection operator (Lasso), which was firstly introduced in Tibshirani (1996), has been one of the most popular approaches because of its benign theoretical guarantee and excellent empirical performance. Recently, a private iteratively hard thresholding pursuit algorithm was developed in Cai et al. (2019) to solve Lasso problem in the differential privacy framework. To deal with the private sign recovery problem in the distributed setup, a naive approach is to solve the private Lasso problem on each local machine and take the signs of the average of all local estimators. However, this method also requires the boundedness assumption on the covariates, and it is too complicated for sign recovery. More importantly, the average of the local Lasso estimators is likely to cause more non-zero elements because the coordinate will become non-zero as long as one of these local estimators is non-zero at this coordinate.

By leveraging the idea of Med-DC, we present Algorithm 2 for sign recovery of sparse regression.

---

**Algorithm 2** Median divide-and-conquer for sparse linear regression (Med-DC Lasso)

---

**Input:** Data on local machines $\{(\boldsymbol{X}_i, Y_i) \mid i \in \mathcal{H}_j\}$ for $j = 1, \ldots, m$, the universal regularization parameter $\lambda_N$.

1: **for** $j = 1, \ldots, m$ **do**
2:     The $j$-th part $\mathcal{H}_j$ computes the local sample mean

$$\hat{\boldsymbol{\theta}}_j = \underset{\boldsymbol{\theta} \in \mathbb{R}^p}{\mathrm{argmin}} \frac{1}{2n} \sum_{i \in \mathcal{H}_j} (Y_i - \boldsymbol{X}_i^{\mathrm{T}} \boldsymbol{\theta})^2 + \lambda_N |\boldsymbol{\theta}|_1. \tag{7}$$

    Then the $j$-th local machine sends $\mathcal{Q}_0(\hat{\boldsymbol{\theta}}_j)$ to the server.
3: **end for**
4: The server takes coordinate-wise median $\hat{\boldsymbol{Q}}(\mathbb{X}) = \mathrm{med}(\mathcal{Q}_0(\hat{\boldsymbol{\theta}}_j) \mid 1 \le j \le m)$.

**Output:** The vector of signs $\hat{\boldsymbol{Q}}(\mathbb{X})$.

---

Similarly as Algorithm 1, every step of the Med-DC Lasso method is deterministic. Therefore, we can solve each local subproblem (7) efficiently by many well-developed algorithms like FISTA in Beck & Teboulle (2009), ADMM in Boyd et al. (2011), etc.

### 3.2 Theory of Regression Parameter Sign Recovery

For linear regression, we consider the following distribution space

$$\mathcal{P}_{\boldsymbol{X}, Y}(\boldsymbol{\theta}^*, \eta_1, C_1, \eta_2, C_2) = \Big\{ \mathbb{P} \Big| \sup_{|\boldsymbol{v}|_2 = 1} \mathbb{E}_{\mathbb{P}} \big\{ \exp(\eta_1 |\boldsymbol{v}^{\mathrm{T}} \boldsymbol{X}|^2) \big\} \le C_1,$$
$$z = Y - \boldsymbol{X}^{\mathrm{T}} \boldsymbol{\theta}^*, z \perp \boldsymbol{X}, \mathbb{E}_{\mathbb{P}} \big\{ \exp(\eta_2 |z|^2) \big\} \le C_2 \Big\}, \tag{8}$$

where $C_1, C_2, \eta_1, \eta_2$ are some positive constants. This implies that both the covariate vector $\boldsymbol{X}$ and the noise $z$ are sub-Gaussian. Then we have the following result of sign recovery consistency.

**Theorem 2.** *(Sign consistency of Med-DC Lasso) Let* $N = mn$ *samples* $\mathbb{X} = \{(\boldsymbol{X}_1, Y_1), \ldots, (\boldsymbol{X}_N, Y_N)\}$ *from* $\mathcal{P}_{\boldsymbol{X}, Y}(\boldsymbol{\theta}^*, \eta_1, C_1, \eta_2, C_2)$ *be evenly distributed in* $m$ *subsets*

$\mathcal{H}_1, \ldots, \mathcal{H}_m$. *Moreover, there are some sufficient large constants $C_3, C_4, \Delta_0 > 0$ such that*

(a) *The dimension $p$ is fixed, and take $\lambda_N = C_3(\sqrt{\log n / N} + \log n/n)$;*

(b) *Denote $S = \text{supp}(\boldsymbol{\mu}^*)$, the minimal signal satisfies $\min_{l \in S} |\mu_l^*| \geq C_4 \lambda_N$;*

(c) *The covariance matrix $\boldsymbol{\Sigma} = \mathbb{E} \boldsymbol{X} \boldsymbol{X}^{\mathrm{T}}$ is positive definite. Let $\boldsymbol{\Sigma}^{-1} = (\boldsymbol{\omega}_1, \ldots, \boldsymbol{\omega}_p)$, there is*

$$\max_{l \in S^c} \frac{|\boldsymbol{\omega}_{-l}|_1}{\omega_{l,l}} \leq 1 - \Delta_0.$$

*Then $\hat{\boldsymbol{Q}}(\mathbb{X})$ defined in Algorithm 2 satisfies that, for some large $\gamma_1$ depends on $C_3, C_4, \Delta_0$, there is*

$$\mathbb{P}\Big(\hat{\boldsymbol{Q}}(\mathbb{X}) = \text{sgn}(\boldsymbol{\theta}^*)\Big) \geq 1 - n^{-\gamma_1}.$$

From the assumption (b), when $m \log n = O(n)$, the minimal signal has the order of $O(\sqrt{\log n / N})$, which meets the "beta-min" condition (Wainwright, 2009) of standard Lasso problem for full sample case (all $N$ samples stored in a single machine). Note that the regularization parameter $\lambda_N$ is a universal constant among all local machines. By assumption (a), the regularization parameter satisfies $\lambda_N \asymp \sqrt{\log n / N}$, which is smaller than the standard setting $O(\sqrt{\log n / n})$ (Lee et al., 2017). Therefore, the local estimators $\hat{\boldsymbol{\theta}}_j$ is not very sparse because the regularization parameter $\lambda_N$ is unable to annihilate the noises brought by the local data. However, by taking the median among the local signs, the noises are canceled out and the signals become detectable. Owing to the smaller scale, it helps to identify the smaller magnitude of signals. The assumption (c) implies that, for $l \in S^c$, the $l$-th row of the precision machine $\boldsymbol{\Sigma}^{-1}$ is dominated by the diagonal entry $\omega_{l,l}$. It can be regarded as a more strict irrepresentability condition (Zhao & Yu, 2006; Wainwright, 2009).

**Proposition 2.** *(Privacy of Med-DC Lasso) Under the same assumptions as Theorem 2, denote $\mathcal{D}(\mathbb{X})$ as the collection of datasets $\mathbb{X}'$ adjacent to $\mathbb{X}$, then for some large $\gamma_2 > 0$, there is*

$$\mathbb{P}\Big(\hat{\boldsymbol{Q}}(\mathbb{X}) = \hat{\boldsymbol{Q}}(\mathbb{X}'), \text{ for all } \mathbb{X}' \in \mathcal{D}(\mathbb{X})\Big) \geq 1 - n^{-\gamma_2}. \tag{9}$$

Similar as (5), equation (9) is a weakened privacy guarantee, which can protect data privacy with high probability. In addition, follow the proofs of Proposition 1 and 2, our proposed methods also guarantee group privacy with high probability (Section 10.1 in Dwork & Roth (2014)).

**Corollary 1.** *(Group Privacy) Under the same assumptions as Theorem 2, denote $\mathcal{D}_k(\mathbb{X})$ as the collection of datasets $\mathbb{X}'$ have at most $k$ elements differing with $\mathbb{X}$, then for some large $\gamma_3 > 0$, then*

$$\mathbb{P}\Big(\hat{\boldsymbol{Q}}(\mathbb{X}) = \hat{\boldsymbol{Q}}(\mathbb{X}'), \text{ for all } \mathbb{X}' \in \mathcal{D}_k(\mathbb{X})\Big) \geq 1 - n^{-\gamma_3}.$$

## 4 SIMULATION STUDY

### 4.1 RESULTS FOR SPARSE MEAN ESTIMATION

In the first experiments, we consider the sparse mean estimation problem, observations $\{\boldsymbol{X}_1, \ldots, \boldsymbol{X}_N\}$ are sampled from the model

$$\boldsymbol{X}_i = \boldsymbol{\mu}^* + \boldsymbol{z}_i,$$

where the noises $\boldsymbol{z}_i$'s are drawn from the multivariate normal distribution $\mathcal{N}(\boldsymbol{0}, \boldsymbol{I}_p)$. We fix the dimension $p$ as 200. The parameter of interest is defined as

$$\boldsymbol{\mu}^* = (1, 0.8, \cdots, 0.2, 0, -0.2, \cdots, -0.8, -1, \boldsymbol{0}_{p-11}^{\mathrm{T}})^{\mathrm{T}}, \tag{10}$$

which means the sparsity level $s$ is fixed as 10. The data is divided into 100 local machines $\mathcal{H}_1, \ldots \mathcal{H}_{100}$, each local sample size is $n = 200$. Therefore, the entire sample size is $N = 200 \times 100$. For the choice of regularization parameter $\lambda_N$ in each local machine, we first choose $\lambda_N'$ based on the dataset in the first local machine $\mathcal{H}_1$ by five-fold cross-validation. Moreover, motivated by the theoretical scale difference in Theorem 1, we further divide $\lambda_n$ by $\sqrt{m}$, namely, $\lambda_N = \lambda_n / \sqrt{m}$. We compare with the following three methods:

(a) **Mean-DC**: Replace the aggregator median in Med-DC by taking average;

(b) **CWZ**: Perform the method proposed in Cai et al. (2019) on all samples in a single machine.

Same as Cai et al. (2019), we adopt the oracle $T = 2\sqrt{\log N}$, $s = 10$, and $(\epsilon, \delta) = (0.5, 10/N^{1.1})$;
    (c) **Pooled-Mean**: Take average among all samples and take quantization function $\mathcal{Q}_{\lambda_N}(\bar{\boldsymbol{X}})$.
Note that **Mean-DC** and **CWZ** need to transmit the local estimators to the server, which is more communication costly. The performance of sign recovery is measured by the following four criteria:

- **Positive and Negative False Discovery Rate.**

$$\text{PFDR} = \frac{\sum_{l \notin S^+}^{p} \mathbb{I}\{\hat{Q}_l(\mathbb{X}) = 1\}}{\max[\sum_{l=1}^{p} \mathbb{I}\{\hat{Q}_l(\mathbb{X}) = 1\}, 1]}, \quad \text{NFDR} = \frac{\sum_{l \notin S^-}^{p} \mathbb{I}\{\hat{Q}_l(\mathbb{X}) = -1\}}{\max[\sum_{l=1}^{p} \mathbb{I}\{\hat{Q}_l(\mathbb{X}) = -1\}, 1]}.$$

- **Total False Discovery Rate and Power.**

$$\text{FDR} = \frac{\sum_{l \notin S}^{p} \mathbb{I}\{\hat{Q}_l(\mathbb{X}) \neq 0\}}{\max[\sum_{l=1}^{p} \mathbb{I}\{\hat{Q}_l(\mathbb{X}) \neq 0\}, 1]}, \quad \text{Power} = \frac{\sum_{l \in S}^{p} \mathbb{I}\{\hat{Q}_l(\mathbb{X}) \neq 0\}}{\max[\sum_{l=1}^{p} \mathbb{I}\{\hat{Q}_l(\mathbb{X}) \neq 0\}, 1]}.$$

Table 1: The PFDR, NFDR, FDR, power, and their standard errors (in parentheses) of different methods under sample size $N = 200 \times 100$, local sample size $n = 200$.

|  | Med-DC | Mean-DC | CWZ | Pooled-Mean |
|---|---|---|---|---|
| PFDR | 0.0684 (0.1251) | 0.9472 (0.0038) | 0.6981 (0.2122) | 0.0000 (0.0000) |
| NFDR | 0.0729 (0.1283) | 0.9473 (0.0036) | 0.6834 (0.2095) | 0.0000 (0.0000) |
| FDR | 0.0751 (0.1170) | 0.9475 (0.0008) | 0.6800 (0.1807) | 0.0000 (0.0000) |
| Power | 1.0000 (0.0000) | 1.0000 (0.0000) | 0.3200 (0.1807) | 0.7840 (0.0846) |

From Table 1, our Med-DC approach clearly outperforms the non-private Mean-DC and private CWZ method. Comparing with the pooled-mean estimator, our method has higher power.

## 4.2 Results for Sparse Linear Regression

In the second experiments, we consider the linear model defined in (6). Let the noises are i.i.d. from $\mathcal{N}(0, 1)$ and assume the i.i.d. covariate vectors $\boldsymbol{X}_i^{\mathrm{T}} = (X_{i,1}, \ldots, X_{i,p})$ $(i = 1, \ldots, N)$ are drawn from a multivariate normal distribution $\mathcal{N}(\boldsymbol{0}, \boldsymbol{\Sigma})$. The covariance matrix $\boldsymbol{\Sigma}$ is a $p \times p$ Toeplitz matrix with its $(i, j)$-th entry $\Sigma_{ij} = 0.5^{|i-j|}$, where $1 \leq i, j \leq p$. We fix the dimension $p = 200$. Moreover, we set the true coefficient $\boldsymbol{\theta}^*$ the same as $\boldsymbol{\mu}^*$ in (10). Similarly, we set $m = 100$, $n = 200$. For the choice of regularization parameter $\lambda_N$ in each local machine, we first choose $\lambda'_N$ based on the dataset in the first local machine $\mathcal{H}_1$ by five-fold cross-validation. As suggested in classical literatures (Wainwright, 2009), motivated by the theoretical scale difference in Theorem 2, and further divide it by $\sqrt{m}$. In addition to **Mean-DC**, we mainly compare with other two methods:

    (d) **CSL**: Use the Communication-efficient Surrogate Likelihood (CSL) framework in Jordan et al. (2019) to obtain an estimator $\hat{\boldsymbol{\theta}}_{CSL}$ for the true parameter and take the signs of it.
    (e) **Pooled-Lasso**: Solve the Lasso problem on all data in a single machine and take the signs.

Note that all the above-mentioned methods are not private. Both the Mean-DC method and the CSL method require to transmit information of local parameters or multi-round gradients, which is more communication-costly than ours. In particular, we note that the CSL method includes an iterative refinement of the estimator. In our simulation study, we present the results of the five-step CSL method. For each experiment, we repeat 500 independent simulations and report the number of PFDR, NFDR, FDR, and power.

Table 2: The PFDR, NFDR, FDR, power and their standard errors (in parentheses) of different methods under sample size $N = 200 \times 100$, local sample size $n = 200$.

|  | Med-DC | Mean-DC | CSL | Pooled-Lasso |
|---|---|---|---|---|
| PFDR | 0.0639 (0.1329) | 0.9454 (0.0425) | 0.4453 (0.3447) | 0.5703 (0.1758) |
| NFDR | 0.0582 (0.1199) | 0.9456 (0.0425) | 0.4325 (0.3480) | 0.5808 (0.1717) |
| FDR | 0.0645 (0.1184) | 0.9457 (0.0424) | 0.4466 (0.3397) | 0.5896 (0.1464) |
| Power | 1.0000 (0.0000) | 1.0000 (0.0000) | 0.9998 (0.0045) | 1.0000 (0.0000) |

It can be observed from Table 2, while all these methods can select the true support set, the Med-DC Lasso method has apparently smaller false discovery rate than others.

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

## A  TECHNICAL LEMMAS

**Lemma 1.** *(Berry-Esseen Theorem, Theorem 9.1.3 in Chow & Teicher (2012)) If $\{X_i,\ i \geq 1\}$ are i.i.d. mean-zero random variables with $\mathbb{E}[X_i^2] = \sigma^2, \mathbb{E}|X_i|^3 < \infty$. Then there exists a constant $C_{\mathrm{B}} > 0$ such that*

$$\sup_{-\infty < x < \infty} \left| \mathbb{P}\left( \sum_{i=1}^{n} X_i < \sqrt{n}\sigma x \right) - \Phi(x) \right| \leq \frac{C_{\mathrm{B}}}{\sqrt{n}}.$$

**Lemma 2.** *(Exponential Inequality, Lemma 1 in Cai & Liu (2011)) Let $X_1, ..., X_n$ be i.i.d. random variables with zero mean. Suppose that there exist some $\eta > 0$ and $C > 0$ such that $\mathbb{E}[X_i^2 e^{\eta|X_i|}] \leq C$. Then uniformly for $0 < x \leq C$ and $n \geq 1$, there is*

$$\mathbb{P}\left\{ \frac{1}{n}\sum_{i=1}^{n} X_i \geq (\eta + \eta^{-1})x \right\} \leq \exp\left( -\frac{nx^2}{C} \right).$$

**Lemma 3.** *Let $N(= mn)$ i.i.d. random variables $X_1, ..., X_N$ evenly distributed in $m$ subsets $\mathcal{H}_1, ..., \mathcal{H}_m$. Suppose $\mathbb{E}[X_i] = 0, \mathbb{E}[X_i^2] = \sigma^2, \mathbb{E}|X_i|^3 < \infty$. Denote $\bar{X}_j = \sum_{i \in \mathcal{H}_j} X_i/n$ as the local sample mean on $\mathcal{H}_j$. For every $\gamma > 1$, denote*

$$c_\gamma = \tilde{C}\left( \frac{1}{n} + \frac{k}{m\sqrt{n}} + \sqrt{\frac{\gamma \log n}{mn}} \right),$$

*where $\tilde{C} > 0$ is sufficiently large enough. Then for every fixed non-negative constant $k$, there is*

$$\mathbb{P}\left\{ \sum_{j=1}^{m} \mathbb{I}\left(\bar{X}_j < c_\gamma\right) \leq \frac{m}{2} + k \right\} + \mathbb{P}\left\{ \sum_{j=1}^{m} \mathbb{I}\left(\bar{X}_j > -c_\gamma\right) \leq \frac{m}{2} + k \right\} = O(n^{-\gamma}). \tag{11}$$

*Proof.* For every $x > 0$, there is

$$\mathbb{P}\left\{ \sum_{j=1}^{m} \mathbb{I}\left(\bar{X}_j < x\right) \leq \frac{m}{2} + k \right\}$$

$$= \mathbb{P}\left\{ \frac{1}{m}\sum_{j=1}^{m} \mathbb{I}\left(\bar{X}_j < x\right) - \mathbb{P}\left(\bar{X}_1 < x\right) \leq \frac{1}{2} + \frac{k}{m} - \mathbb{P}\left(\bar{X}_1 < x\right) \right\}$$

$$\leq \mathbb{P}\left\{ \frac{1}{m}\sum_{j=1}^{m} \mathbb{I}\left(\bar{X}_j < x\right) - \mathbb{P}\left(\bar{X}_1 < x\right) \leq \frac{1}{2} + \frac{k}{m} + \frac{C_{\mathrm{B}}}{\sqrt{n}} - \Phi\left(\frac{\sqrt{n}x}{\sigma}\right) \right\}$$

$$= \mathbb{P}\left\{ \frac{1}{m}\sum_{j=1}^{m} \mathbb{I}\left(\bar{X}_j < x\right) - \mathbb{P}\left(\bar{X}_1 < x\right) \leq -\sqrt{\frac{C\gamma \log n}{m}} \right\},$$

where the last line uses Berry-Esseen Theorem (Lemma 1), and $x$ is given by

$$x = \frac{\sigma}{\sqrt{n}}\Phi^{-1}\left( \frac{1}{2} + \frac{k}{m} + \frac{C_{\mathrm{B}}}{\sqrt{n}} + \sqrt{\frac{C\gamma \log n}{m}} \right).$$

Applying Lemma 2 to the i.i.d. sequence $\mathbb{I}\left(\bar{X}_j < x\right) - \mathbb{P}\left(\bar{X}_1 < x\right)$, we have

$$\mathbb{P}\left\{ \frac{1}{m}\sum_{j=1}^{m} \mathbb{I}\left(\bar{X}_j < x\right) - \mathbb{P}\left(\bar{X}_1 < x\right) \leq -\sqrt{\frac{C\gamma \log n}{m}} \right\} = O(n^{-\gamma}),$$

for some $C$ large enough. Moreover, we have the following elementary facts

$$\left|\Phi^{-1}(x_0)\right| = \left|\Phi^{-1}(x_0) - \Phi^{-1}(1/2)\right| \leq |x_0 - 1/2|\left(\Phi^{-1}\right)'(x_0) \leq \frac{|x_0 - 1/2|}{\psi\left\{\Phi^{-1}(3/4)\right\}},$$

holds for any $1/4 \leq x_0 < 3/4$. On the other hand, we know that $1/4 \leq \Phi(\sqrt{n}x/\sigma) < 3/4$ holds for $m, n$ sufficiently large. Denote $C_\delta = 1/\psi\{\Phi^{-1}(3/4)\}$, then there is

$$\mathbb{P}\left\{\sum_{j=1}^m \mathbb{I}\left(\bar{X}_j < \frac{C_\delta \sigma}{\sqrt{n}}\left(\frac{C_B}{\sqrt{n}} + \frac{k}{m} + \sqrt{\frac{C\gamma \log n}{m}}\right)\right) \leq \frac{m}{2}\right\}$$

$$\leq \mathbb{P}\left\{\frac{1}{m}\sum_{j=1}^m \mathbb{I}\left(\bar{X}_j < x\right) - \mathbb{P}\left(\bar{X}_1 < x\right) \leq -\sqrt{\frac{C\gamma \log n}{m}}\right\} \leq n^{-\gamma}.$$

Therefore, if we choose $\tilde{C} \geq \max\{C_\delta C_B \sigma, C_\delta \sqrt{C}\sigma\}$, the bound of the first term in the left hand side of (11) is proved. By repeating the same procedure, we can prove the bound of the second term, which yields the desired result. $\qquad\square$

**Lemma 4.** *Let $\boldsymbol{X}_1, \ldots, \boldsymbol{X}_n$ be i.i.d. random vectors sampled from the distribution in (8). Denote its covariance matrix as $\boldsymbol{\Sigma}$, and the sample covariance matrix as $\hat{\boldsymbol{\Sigma}}$. Then for every $\gamma > 1$, there exists a constant $\tilde{C} > 0$ such that*

$$\mathbb{P}\left(\left\|\boldsymbol{\Sigma}^{-1}\hat{\boldsymbol{\Sigma}} - \boldsymbol{I}\right\|_\infty \geq \tilde{C}\sqrt{\frac{\log n}{n}}\right) = O(n^{-\gamma}).$$

*Proof.* Recall that the inverse covariance matrix is denoted as $\boldsymbol{\Sigma}^{-1} = (\boldsymbol{\omega}_1, \ldots, \boldsymbol{\omega}_p)$, and $\boldsymbol{e}_l$ as the $l$-th coordinate vector. Then the $(l_1, l_2)$-entry of the matrix $\boldsymbol{\Sigma}^{-1}\hat{\boldsymbol{\Sigma}} - \boldsymbol{I}$ is

$$(\boldsymbol{\Sigma}^{-1}\hat{\boldsymbol{\Sigma}} - \boldsymbol{I})_{l_1, l_2} = \frac{1}{n}\sum_{i=1}^n \boldsymbol{\omega}_{l_1}^T \boldsymbol{X}_i \cdot \boldsymbol{e}_{l_2}^T \boldsymbol{X}_i - \delta_{l_1, l_2}.$$

Since the dimension $p$ is assumed to be bounded, and the covariance matrix is positive definite, there exist a constant $\rho \in (0, 1)$ such that

$$\rho \leq \Lambda_{\min}(\boldsymbol{\Sigma}) \leq \Lambda_{\max}(\boldsymbol{\Sigma}) \leq \rho^{-1}.$$

Then we have

$$\max_{1 \leq l \leq p} |\boldsymbol{\omega}_l|_2 \leq \left\|\boldsymbol{\Sigma}^{-1}\right\| \leq \rho^{-1}.$$

Since $\boldsymbol{X}$ is sub-Gaussain in (8), we obtain

$$\max_{1 \leq l_1, l_2 \leq p} \mathbb{E}\left\{\exp \eta_1 \rho \left|\boldsymbol{\omega}_{l_1}^T \boldsymbol{X}_i \cdot \boldsymbol{e}_{l_2}^T \boldsymbol{X}_i - \delta_{l_1, l_2}\right|\right\}$$

$$\leq e^{\eta_1 \rho} \cdot \sup_{|\boldsymbol{v}|_2 \leq 1} \mathbb{E}\left\{\eta_1 |\boldsymbol{v}^T \boldsymbol{X}|^2\right\} \leq e^{\eta_1 \rho} C_1.$$

Therefore, we can apply Lemma 2 to each coordinate and yield

$$\mathbb{P}\left(\left\|\boldsymbol{\Sigma}^{-1}\hat{\boldsymbol{\Sigma}} - \boldsymbol{I}\right\|_\infty \geq \tilde{C}\sqrt{\frac{\log n}{n}}\right)$$

$$\leq p^2 \max_{1 \leq l_1, l_2 \leq p} \mathbb{P}\left(\left|\frac{1}{n}\sum_{i=1}^n \boldsymbol{\omega}_{l_1}^T \boldsymbol{X}_i \cdot \boldsymbol{e}_{l_2}^T \boldsymbol{X}_i - \delta_{l_1, l_2}\right| \geq \tilde{C}\sqrt{\frac{\log n}{n}}\right)$$

$$= O(p^2 n^{-\gamma-2}) = O(n^{-\gamma}),$$

for some $\tilde{C}$ sufficiently large. Therefore, the lemma is proved. $\qquad\square$

# B    PROOF OF MAIN RESULTS

*Proof of Theorem 1.* If we can show that

$$\max_{1 \leq l \leq p} \mathbb{P}\left(\hat{Q}_l(\mathbb{X}) \neq \operatorname{sgn}(\mu_l^*)\right) = O(n^{-\gamma}), \tag{12}$$

where $\gamma > 0$ is large enough. Then this theorem is proved as follows

$$1 - \mathbb{P}\Big(\hat{\boldsymbol{Q}}(\mathbb{X}) = \mathrm{sgn}(\boldsymbol{\mu}^*)\Big)$$

$$\leq p \max_{1 \leq l \leq p} \mathbb{P}\Big(\hat{Q}_l(\mathbb{X}) \neq \mathrm{sgn}(\mu_l^*)\Big) = O(pn^{-\gamma}) = O(n^{-\gamma + \gamma_0}),$$

provided that $\gamma > \gamma_0$. For the $l$-th coordinate, we firstly suppose that $\mu_l^* = 0$. Since $\boldsymbol{X}_i$'s are sampled from the distribution (4), by Lemma 3 with $k = 0$, if

$$\lambda_N = C_0 \left( \sqrt{\frac{\log n}{mn}} + \frac{1}{n} \right) \geq c_\gamma,$$

we have

$$\mathbb{P}\Big(\hat{Q}_l(\mathbb{X}) \neq \mathrm{sgn}(\mu_l^*)\Big)$$

$$\leq \mathbb{P}\Big\{ \sum_{j=1}^m \mathbb{I}\Big( \bar{X}_{j,l} < \lambda_N \Big) \leq \frac{m}{2} \Big\} + \mathbb{P}\Big\{ \sum_{j=1}^m \mathbb{I}\Big( \bar{X}_{j,l} > -\lambda_N \Big) \leq \frac{m}{2} \Big\} = O(n^{-\gamma}).$$

Next we assume $\mu_l^* > 0$. We use Lemma 3 again, if

$$\mu_l^* \geq C_1 \Big( \sqrt{\frac{\log n}{mn}} + \frac{1}{n} \Big) \geq c_\gamma + \lambda_N,$$

then there is

$$\mathbb{P}\Big(\hat{Q}_l(\mathbb{X}) \neq \mathrm{sgn}(\mu_l^*)\Big) \leq \mathbb{P}\Big\{ \sum_{j=1}^m \mathbb{I}\Big( \bar{X}_{j,l} > \lambda_N \Big) \leq \frac{m}{2} \Big\}$$

$$\leq \mathbb{P}\Big\{ \sum_{j=1}^m \mathbb{I}\Big( \bar{X}_{j,l} - \mu_l^* > -c_\gamma \Big) \leq \frac{m}{2} \Big\} = O(n^{-\gamma}).$$

Lastly, when $\mu_l^* < 0$, the proof is the same as above. Therefore (12) is proved. $\qquad\square$

*Proof of Proposition 1.* For $\mathbb{X}' \in \mathcal{D}(\mathbb{X})$, denote the elements in $\mathbb{X}'$ as $\boldsymbol{X}_i'$ (where $1 \leq i \leq N$), where

$$\sum_{i=1}^N \mathbb{I}(\boldsymbol{X}_i' \neq \boldsymbol{X}_i) = 1.$$

Moreover, since these data are stored in $m$ different machines $\mathcal{H}_1, \ldots, \mathcal{H}_m$, we have

$$0 \leq \sum_{j=1}^m \mathbb{I}(\bar{\boldsymbol{X}}_j' \neq \bar{\boldsymbol{X}}_j) \leq 1. \tag{13}$$

Noticing that

$$\mathbb{P}\Big(\hat{\boldsymbol{Q}}(\mathbb{X}) \neq \hat{\boldsymbol{Q}}(\mathbb{X}'), \text{ for all } \mathbb{X}' \in \mathcal{D}(\mathbb{X})\Big)$$

$$\leq \mathbb{P}\Big(\hat{\boldsymbol{Q}}(\mathbb{X}') \neq \mathrm{sgn}(\boldsymbol{\mu}^*), \text{ for all } \mathbb{X}' \in \mathcal{D}(\mathbb{X}); \hat{\boldsymbol{Q}}(\mathbb{X}) = \mathrm{sgn}(\boldsymbol{\mu}^*)\Big) + \mathbb{P}\Big(\hat{\boldsymbol{Q}}(\mathbb{X}) \neq \mathrm{sgn}(\boldsymbol{\mu}^*)\Big)$$

$$\leq \mathbb{P}\Big(\hat{\boldsymbol{Q}}(\mathbb{X}') \neq \mathrm{sgn}(\boldsymbol{\mu}^*), \text{ for all } \mathbb{X}' \in \mathcal{D}(\mathbb{X})\Big) + O(n^{-\gamma_1}).$$

Therefore, we only need to show that

$$\mathbb{P}\Big(\hat{\boldsymbol{Q}}(\mathbb{X}') \neq \mathrm{sgn}(\boldsymbol{\mu}^*), \text{ for all } \mathbb{X}' \in \mathcal{D}(\mathbb{X})\Big) = O(n^{-\gamma_2}). \tag{14}$$

For the $l$-th coordinate, we firstly suppose that $\mu_l^* = 0$. Then there is

$$\mathbb{P}\Big(\hat{Q}_l(\mathbb{X}') \neq 0, \text{ for all } \mathbb{X}' \in \mathcal{D}(\mathbb{X})\Big)$$

$$\leq \mathbb{P}\Big\{ \sum_{j=1}^m \mathbb{I}\Big(\bar{X}_{j,l}' < \lambda_N\Big) \leq \frac{m}{2}, \text{ for all } \mathbb{X}' \in \mathcal{D}(\mathbb{X})\Big\}$$

$$+ \mathbb{P}\Big\{ \sum_{j=1}^m \mathbb{I}\Big(\bar{X}_{j,l}' > -\lambda_N\Big) \leq \frac{m}{2}, \text{ for all } \mathbb{X}' \in \mathcal{D}(\mathbb{X})\Big\}$$

$$\leq \mathbb{P}\Big\{ \sum_{j=1}^m \mathbb{I}\Big(\bar{X}_{j,l} < \lambda_N\Big) \leq \frac{m}{2} + 1\Big\} + \mathbb{P}\Big\{ \sum_{j=1}^m \mathbb{I}\Big(\bar{X}_{j,l} > -\lambda_N\Big) \leq \frac{m}{2} + 1\Big\},$$

where the last inequality uses (13). Using Lemma 3 with $k = 1$, with $\lambda_N$ properly chosen, there is

$$\mathbb{P}\Big\{ \sum_{j=1}^m \mathbb{I}\Big(\bar{X}_{j,l} < \lambda_N\Big) \leq \frac{m}{2} + 1\Big\} + \mathbb{P}\Big\{ \sum_{j=1}^m \mathbb{I}\Big(\bar{X}_{j,l} > -\lambda_N\Big) \leq \frac{m}{2} + 1\Big\} = O(n^{-\gamma}),$$

for some $\gamma > 0$. Similarly, if $\mu_l^* > 0$, there is

$$\mathbb{P}\Big(\hat{Q}_l(\mathbb{X}') \neq 1, \text{ for all } \mathbb{X}' \in \mathcal{D}(\mathbb{X})\Big)$$

$$\leq \mathbb{P}\Big\{ \sum_{j=1}^m \mathbb{I}\Big(\bar{X}_{j,l}' > \lambda_N\Big) \leq \frac{m}{2}, \text{ for all } \mathbb{X}' \in \mathcal{D}(\mathbb{X})\Big\}$$

$$\leq \mathbb{P}\Big\{ \sum_{j=1}^m \mathbb{I}\Big(\bar{X}_{j,l} > \lambda_N\Big) \leq \frac{m}{2} + 1\Big\}$$

$$\leq \mathbb{P}\Big\{ \sum_{j=1}^m \mathbb{I}\Big(\bar{X}_{j,l} - \mu_l^* > \lambda_N - \mu_l^*\Big) \leq \frac{m}{2} + 1\Big\} = O(n^{-\gamma}),$$

where the penultimate line uses (13) and the last line uses Lemma 3 with $k = 1$. The proof when $\mu_l^* < 0$ is similar, therefore

$$\mathbb{P}\Big(\hat{Q}(\mathbb{X}') \neq \operatorname{sgn}(\boldsymbol{\mu}^*), \text{ for all } \mathbb{X}' \in \mathcal{D}(\mathbb{X})\Big)$$

$$\leq p \mathbb{P}\Big(\hat{Q}_l(\mathbb{X}') \neq \operatorname{sgn}(\mu_l^*), \text{ for all } \mathbb{X}' \in \mathcal{D}(\mathbb{X})\Big) = O(pn^{-\gamma}) = O(n^{-\gamma_2}),$$

which proves (14). Therefore the proposition is proved. $\qquad\square$

*Proof of Theorem 2.* For each $j \in \{1, \ldots, m\}$, taking sub-gradient of (7) at $\hat{\boldsymbol{\theta}}_j$, we have that

$$\frac{1}{n} \sum_{i \in \mathcal{H}_j} (Y_i - \boldsymbol{X}_i^{\mathrm{T}} \hat{\boldsymbol{\theta}}_j) \boldsymbol{X}_i + \lambda_N \boldsymbol{Z}_j = 0,$$

where $\boldsymbol{Z}_j$ is the sub-gradient satisfying $|\boldsymbol{Z}_j|_\infty \leq 1$. Rearranging the terms and multiplying $\boldsymbol{\Sigma}^{-1}$ on the both sides, we have

$$\hat{\boldsymbol{\theta}}_j - \boldsymbol{\theta}^* = (\boldsymbol{I} - \boldsymbol{\Sigma}^{-1} \frac{1}{n} \sum_{i \in \mathcal{H}_j} \boldsymbol{X}_i \boldsymbol{X}_i^{\mathrm{T}})(\hat{\boldsymbol{\theta}}_j - \boldsymbol{\theta}^*) + \frac{1}{n} \sum_{i \in \mathcal{H}_j} \boldsymbol{\Sigma}^{-1} \boldsymbol{X}_i z_i + \lambda_N \boldsymbol{\Sigma}^{-1} \boldsymbol{Z}_j. \qquad (15)$$

Taking $\tilde{\eta} = \min\{\eta_1 \rho, \eta_2\}$, since the noise and covariates are assumed to be sub-Gaussian in (8), for each coordinate $l \in \{1, \ldots, p\}$, we have

$$\max_{1 \leq l \leq p} \mathbb{E}\big\{ \exp \tilde{\eta} |\boldsymbol{\omega}_l \boldsymbol{X} \cdot z| \big\}$$

$$\leq \max_{1 \leq l \leq p} \mathbb{E}\Big\{ \exp \Big(\frac{1}{2} \eta \rho |\boldsymbol{\omega}_l \boldsymbol{X}|^2 + \frac{1}{2} \eta_2 |z|^2\Big) \Big\}$$

$$\leq \max_{1 \leq l \leq p} \Big[ \mathbb{E}\big\{ \exp \eta \rho |\boldsymbol{\omega}_l \boldsymbol{X}|^2 \big\} \cdot \mathbb{E}\big\{ \exp \eta_2 |z|^2 \big\} \Big]^{1/2} \leq \sqrt{C_1 C_2}.$$

Therefore by Lemma 2, we know that there exists a constant $\tilde{C}_1 > 0$ such that

$$\max_{1 \leq j \leq m} \Big| \frac{1}{n} \sum_{i \in \mathcal{H}_j} \mathbf{\Sigma}^{-1} \mathbf{X}_i z_i \Big|_\infty \leq \tilde{C}_1 \sqrt{\frac{\log n}{n}}, \tag{16}$$

with probability larger than $1 - O(n^{-\gamma})$. On the other hand, by the fact that $|\mathbf{Z}_j|_\infty \leq 1$, we have

$$|\lambda_N \mathbf{\Sigma}^{-1} \mathbf{Z}_j|_\infty \leq \lambda_N \left\| \mathbf{\Sigma}^{-1} \right\|_{L_\infty}. \tag{17}$$

Moreover, by Lemma 4, we know

$$\max_{1 \leq j \leq m} \Big\| \mathbf{I} - \mathbf{\Sigma}^{-1} \frac{1}{n} \sum_{i \in \mathcal{H}_j} \mathbf{X}_i \mathbf{X}_i^{\mathrm{T}} \Big\|_{L_\infty} = O_{\mathbb{P}} \Big( \sqrt{\frac{\log n}{n}} \Big) \tag{18}$$

Substitute (16) (17) (18) into (15), we have

$$|\hat{\boldsymbol{\theta}}_j - \boldsymbol{\theta}^*|_\infty \leq 2\lambda_N \left\| \mathbf{\Sigma}^{-1} \right\|_{L_\infty} + 2\tilde{C}_1 \sqrt{\frac{\log n}{n}}.$$

From (15), the $l$-th coordinate can be written in the following form

$$\hat{\theta}_{j,l} - \theta_l^* = \lambda_N \omega_{l,l} Z_{j,l} + \lambda_N \omega_{l,-l} Z_{j,-l} + \frac{1}{n} \sum_{i \in \mathcal{H}_j} \boldsymbol{\omega}_l^{\mathrm{T}} \mathbf{X}_i z_i + o_{\mathbb{P}} \Big( \frac{\log n}{n} \Big). \tag{19}$$

It left to rehash the argument in the proof of Theorem 1. From Lemma 5 below we have that

$$1 - \mathbb{P} \Big( \hat{\boldsymbol{Q}}(\mathbb{X}) = \mathrm{sgn}(\boldsymbol{\theta}^*) \Big) \leq p \cdot \max_{1 \leq l \leq p} \mathbb{P} \Big( \hat{Q}_l(\mathbb{X}) \neq \mathrm{sgn}(\theta_l^*) \Big) = O(n^{-\gamma+1}).$$

Therefore we proved Theorem 2. $\qquad \square$

**Lemma 5.** *Assume the same assumptions in Theorem 2. For every $1 \leq l \leq p$, we have*

$$\mathbb{P} \Big( \hat{Q}_l(\mathbb{X}) \neq \mathrm{sgn}(\theta_l^*) \Big) = O(n^{-\gamma}),$$

*for arbitrarily fixed $\gamma > 1$.*

*Proof.* When $\theta_l^* = 0$, we know that

$$\mathbb{P} \Big( \hat{Q}_l(\mathbb{X}) \neq \mathrm{sgn}(\theta_l^*) \Big) \leq \mathbb{P} \Big\{ \sum_{j=1}^m \mathbb{I} \Big( \hat{\theta}_{j,l} \leq 0 \Big) \leq \frac{m}{2} \Big\} + \mathbb{P} \Big\{ \sum_{j=1}^m \mathbb{I} \Big( \hat{\theta}_{j,l} \geq 0 \Big) \leq \frac{m}{2} \Big\}.$$

By symmetry of the formulation, it is enough to prove

$$\mathbb{P} \Big\{ \sum_{j=1}^m \mathbb{I} \Big( \hat{\theta}_{j,l} > 0 \Big) \geq \frac{m}{2} \Big\} = O(n^{-\gamma}). \tag{20}$$

When $\hat{\theta}_{j,l} > 0$, we know that $Z_{j,l} = 1$ (see (19)). Moreover, by assumption (c), we have

$$\lambda_N \omega_{l,l} Z_{j,l} + \lambda_N \omega_{l,-l} Z_{j,-l} \geq \lambda_N \omega_{l,l} - \lambda_N |\omega_{l,-l}|_1 \geq \Delta_0 \lambda_N \omega_{l,l}.$$

Therefore from (19) we have that

$$\mathbb{P} \Big\{ \sum_{j=1}^m \mathbb{I} \Big( \hat{\theta}_{j,l} > 0 \Big) \geq \frac{m}{2} \Big\} \leq \mathbb{P} \Big\{ \sum_{j=1}^m \mathbb{I} \Big( \frac{1}{n} \sum_{i \in \mathcal{H}_j} \boldsymbol{\omega}_l^{\mathrm{T}} \mathbf{X}_i z_i \leq \frac{1}{2} \Delta_0 \lambda_N \omega_{l,l} \Big) \leq \frac{m}{2} \Big\}.$$

Applying Lemma 3 with $k = 0$ to the i.i.d. random variables $\boldsymbol{\omega}_l^{\mathrm{T}} \mathbf{X}_i z_i$ we can prove (20) by taking

$$\lambda_N = \frac{2\tilde{C}_1}{\Delta_0 \omega_{l,l}} \Big( \sqrt{\frac{\log n}{mn}} + \frac{\log n}{n} \Big), \tag{21}$$

with $\tilde{C}$ sufficiently large. Repeat the argument for the other half, we can prove the case for $\theta_l^* = 0$.

When $\theta_l^* > 0$, again from equation (19), we have

$$\hat{\theta}_{j,l} \geq \theta_l^* + \frac{1}{n} \sum_{i \in \mathcal{H}_j} \boldsymbol{\omega}_l^{\mathrm{T}} \boldsymbol{X}_i z_i - \left\| \boldsymbol{\Sigma}^{-1} \right\|_{L_\infty} \lambda_N + o_{\mathbb{P}}\Big(\frac{\log n}{n}\Big).$$

Therefore

$$\mathbb{P}\Big(\hat{Q}_l(\mathbb{X}) \neq \mathrm{sgn}(\theta_l^*)\Big) \leq \mathbb{P}\Big\{ \sum_{j=1}^m \mathbb{I}\Big(\hat{\theta}_{j,l} \leq 0\Big) \geq \frac{m}{2} \Big\}$$

$$\leq \mathbb{P}\Big\{ \sum_{j=1}^m \mathbb{I}\Big(\frac{1}{n} \sum_{i \in \mathcal{H}_j} \boldsymbol{\omega}_l^{\mathrm{T}} \boldsymbol{X}_i z_i \leq \theta_l^* - 2 \left\| \boldsymbol{\Sigma}^{-1} \right\|_{L_\infty} \lambda_N\Big) \geq \frac{m}{2} \Big\}.$$

Applying Lemma 3 we can show that

$$\mathbb{P}\Big\{ \sum_{j=1}^m \mathbb{I}\Big(\frac{1}{n} \sum_{i \in \mathcal{H}_j} \boldsymbol{\omega}_l^{\mathrm{T}} \boldsymbol{X}_i z_i \leq \theta_l^* - 2 \left\| \boldsymbol{\Sigma}^{-1} \right\|_{L_\infty} \lambda_N\Big) \geq \frac{m}{2} \Big\} = O(n^{-\gamma}),$$

provided that

$$\theta_l^* - 2 \left\| \boldsymbol{\Sigma}^{-1} \right\|_{L_\infty} \lambda_N \geq \tilde{C}_2\Big(\sqrt{\frac{\log n}{mn}} + \frac{\log n}{n}\Big),$$

with $\tilde{C}_2$ sufficiently large. Combining with (21) we know

$$\theta_l^* \geq \tilde{C}_3 \lambda_N,$$

for some $\tilde{C}_3 > 0$. When $\theta_l^* < 0$, the prove is essentially the same as above, hence we omit it for brevity. Thus the lemma is proved. $\qquad\square$

*Proof of Proposition 2.* The proof is similar as that of Proposition 1. For $\mathbb{X}' \in \mathcal{D}(\mathbb{X})$, denote the elements in $\mathbb{X}'$ as $(\boldsymbol{X}_i', Y_i')$ (where $1 \leq i \leq N$), where

$$\sum_{i=1}^N \mathbb{I}\Big((\boldsymbol{X}_i', Y_i) \neq (\boldsymbol{X}_i, Y_i)\Big) = 1.$$

Since these data are stored in $m$ machines, we denote $\hat{\boldsymbol{\theta}}_j'$ as the local estimator given by data $\{(\boldsymbol{X}_i', Y_i') \mid i \in \mathcal{H}_j\}$, then there is

$$0 \leq \sum_{j=1}^m \mathbb{I}(\hat{\boldsymbol{\theta}}_j' \neq \hat{\boldsymbol{\theta}}_j) \leq 1. \tag{22}$$

Noticing that

$$\mathbb{P}\Big(\hat{\boldsymbol{Q}}(\mathbb{X}) \neq \hat{\boldsymbol{Q}}(\mathbb{X}'), \text{ for all } \mathbb{X}' \in \mathcal{D}(\mathbb{X})\Big)$$

$$\leq \mathbb{P}\Big(\hat{\boldsymbol{Q}}(\mathbb{X}') \neq \mathrm{sgn}(\boldsymbol{\theta}^*), \text{ for all } \mathbb{X}' \in \mathcal{D}(\mathbb{X}); \hat{\boldsymbol{Q}}(\mathbb{X}) = \mathrm{sgn}(\boldsymbol{\theta}^*)\Big) + \mathbb{P}\Big(\hat{\boldsymbol{Q}}(\mathbb{X}) \neq \mathrm{sgn}(\boldsymbol{\theta}^*)\Big)$$

$$\leq \mathbb{P}\Big(\hat{\boldsymbol{Q}}(\mathbb{X}') \neq \mathrm{sgn}(\boldsymbol{\theta}^*), \text{ for all } \mathbb{X}' \in \mathcal{D}(\mathbb{X})\Big) + O(n^{-\gamma_1}).$$

Therefore we only need to prove

$$\mathbb{P}\Big(\hat{\boldsymbol{Q}}(\mathbb{X}') \neq \mathrm{sgn}(\boldsymbol{\theta}^*), \text{ for all } \mathbb{X}' \in \mathcal{D}(\mathbb{X})\Big) = O(n^{-\gamma_2}). \tag{23}$$

When $\theta_l^* = 0$, we have

$$\mathbb{P}\Big(\hat{Q}_l(\mathbb{X}') \neq 0, \text{ for all } \mathbb{X}' \in \mathcal{D}(\mathbb{X})\Big)$$

$$\leq \mathbb{P}\Big\{ \sum_{j=1}^m \mathbb{I}\Big(\hat{\theta}_{j,l}' \leq 0\Big) \leq \frac{m}{2}, \text{ for all } \mathbb{X}' \in \mathcal{D}(\mathbb{X}) \Big\}$$

$$+ \mathbb{P}\Big\{ \sum_{j=1}^m \mathbb{I}\Big(\hat{\theta}_{j,l}' \geq 0\Big) \leq \frac{m}{2}, \text{ for all } \mathbb{X}' \in \mathcal{D}(\mathbb{X}) \Big\}$$

$$\leq \mathbb{P}\Big\{ \sum_{j=1}^m \mathbb{I}\Big(\hat{\theta}_{j,l} \leq 0\Big) \leq \frac{m}{2} + 1 \Big\} + \mathbb{P}\Big\{ \sum_{j=1}^m \mathbb{I}\Big(\hat{\theta}_{j,l} \geq 0\Big) \leq \frac{m}{2} + 1 \Big\},$$

where the last line uses (22). Using the expansion (19) and rehash the proof in Lemma 5, we can show that

$$\mathbb{P}\Big\{\sum_{j=1}^{m}\mathbb{I}\Big(\hat{\theta}_{j,l}\leq 0\Big)\leq\frac{m}{2}+1\Big\}+\mathbb{P}\Big\{\sum_{j=1}^{m}\mathbb{I}\Big(\hat{\theta}_{j,l}\geq 0\Big)\leq\frac{m}{2}+1\Big\}=O(n^{-\gamma}).$$

When $\theta_l^* > 0$, similarly we have that

$$\mathbb{P}\Big(\hat{Q}_l(\mathbb{X}')\neq 1,\ \text{for all }\mathbb{X}'\in\mathcal{D}(\mathbb{X})\Big)$$

$$\leq\mathbb{P}\Big\{\sum_{j=1}^{m}\mathbb{I}\Big(\hat{\theta}'_{j,l}\leq 0\Big)\geq\frac{m}{2},\ \text{for all }\mathbb{X}'\in\mathcal{D}(\mathbb{X})\Big\}$$

$$\leq\mathbb{P}\Big\{\sum_{j=1}^{m}\mathbb{I}\Big(\hat{\theta}_{j,l}\leq 0\Big)\geq\frac{m}{2}-1\Big\}=O(n^{-\gamma}).$$

Similar argument can be applied for the case when $\theta_l^* < 0$, which concludes the proof of (23). Therefore, Proposition 2 is proved. $\qquad\square$

