# OpenReview forum: "Median DC for Sign Recovery: Privacy can be Achieved by Deterministic Algorithms"
_ICLR.cc/2021/Conference — Reject_

### Official Review · AnonReviewer4 · 2020-10-14
**Median DC review. Decision: Reject.**

**Rating:** 4
**Confidence:** 5

**Review:**

The paper gives "almost private" algorithms for problem of sign recovery of mean vector and of linear regression. The techniques follow their general framework of Median DC, which is similar to the well-known median-of-means approach. They give theoretical guarantees for the same, along with empirical results comparing with known differentially private algorithms and their non-private counterparts.

Strengths: Strong technical justifications in terms of proofs and experiments.

Weaknesses: I don't totally understand the message of the paper. Why is loss of worst-case privacy acceptable? This is what happens in the paper, and without a good justification for that, I don't see why such a loss of privacy is okay. Even privacy has not been formally defined. Apart from group privacy, no other important properties of the "definition" have been proved.

Nitpicks:
-The title is a bit misleading, even though it is obvious that DP cannot be achieved by deterministic algorithms.
-You talk about certain papers doing mean estimation on page 3, but you forget to mention a recent result by Kamath et al on private mean estimation of heavy-tailed distributions, which also uses median of means framework, and may have connections to robustness.
-At the bottom of page 5, you say that the method is roughly regarded as a (0,0)-DP algorithm. That's a very brave thing to say. I wouldn't claim things like that without formal justifications.

Score justification: The weaknesses of this paper just outweigh the positives. A good motivation/justification of the privacy model could have helped to get a better score.

---

### Official Review · AnonReviewer2 · 2020-10-26
**ICLR 2021 Conference Paper3411 AnonReviewer2**

**Rating:** 4
**Confidence:** 4

**Review:**

Summary:
The paper considers the sign recovery problem in a distributed setting with privacy constraints. The paper proposes an algorithm “median divide-and-conquer (Med-DC)” which takes the sign locally in each machine and then takes the median globally. The paper shows that in the sparse mean estimation setting, Med-DC is correct with high probability under some assumptions and Med-DC satisfies a weaker notion of differential privacy proposed by the paper. The paper then extends this algorithm to the sparse linear regression setting.

Concerns:
The main concern about the paper is whether the “weaker” differential privacy notion proposed by the paper makes sense.

To me, this modification on differential privacy is very big, but it is not well discussed in the paper. In my understanding, the main difference between this notion and the standard differential privacy is that the standard differential privacy considers the worst case of the input but the modified notion considers the average case when inputs are assumed to be sampled from a distribution. From a single user’s perspective, this new privacy notion makes sense only when the user trusts that other users will follow the mechanism.

In this new notion, if you simply take the median of n binary numbers sampled from Bernoulli half, the median (without any noise) is private because with high probability, a single number flip won’t flip the median. And this shows that it might be very easy to design deterministic algorithms under this new privacy notion. It seems to me that the main reason for Med-DC being both deterministic and private in this new notion is the weakness of the new privacy notion but not the well-design of Med-DC.

Reasons for score:
I vote for rejection. As discussed in the concerns, the modification of the differential privacy notion makes the problem very different and this modification is not well justified in the paper.

Typos and minor comments:
(1) Page 2: the definition of the supp(v) is not very clear. It’s written as {condition 1 | condition 2}. What are the elements in the set?
(2) Page 2: when a_n = O(b_n) and b_n = O(a_n), you can write a_n = \Theta(b_n) as the standard notation.
(3) Page 6, first paragraph of section 3.1, “recovery” -> “recover”
(4) I find the name “divide-and-conquer” does not fit the algorithm very well. A divide-and-conquer algorithm breaks down a problem into many simpler sub-problems. The algorithm in the paper has data points partitioned because of the distributed setup of the problem.

---

### Official Review · AnonReviewer3 · 2020-10-27
**Paper on robust algorithms for sign recovery.**

**Rating:** 7
**Confidence:** 3

**Review:**

==== Sumary of the problems considered and paper contribution

This paper studies the problem of sign recovery for sparse mean estimation and sparse linear regression. They show that a median based divide and conquer algorithm has high utility (as measured by power, false discovery rate, and positive and negative false discovery rates) and robustness properties. They discuss the privacy implications of these robustness properties, rigorously showing that the deterministic algorithm they define satisfies random differential privacy, where the probability (over the choice of dataset) of not satisfying privacy tends to 0. They show experimentally that their algorithm has low communication and performs well, even when compared to algorithms with no robustness properties.

==== Comments

The paper touches on some interesting questions around robustness and privacy. They essentially design a robust algorithm, subdividing the data to produce averages, reducing the signal by reporting on the sign of each answer, then using the median to give the final answer. Their privacy guarantee amounts to showing that this process is very stable to outliers. The authors remark at one point that this stability implies that they could likely use a procedure like propose test release to give an actually differentially private version. I wonder why they didn’t do this? I’d be very interested to see how it performed.

I think the strength of the stability statements Proposition 1, Proposition 2 and corollary 1 gets a bit lost in the vagueness of the wording. It should be made clear that the randomness is over X, and X’ is any worst case neighbouring dataset. This is significantly stronger than if the randomness was over the pair. This stronger version means that the algorithm is stable against outliers, including those caused by unclean data, or malicious participants. This is particularly interesting in Corollary 1, which discusses group privacy. Also “random differential privacy”, which has appeared in the literature previously, seems like the notion the authors are looking for. However, it seems like a little bit of a stretch to call this "roughly (0,0)-DP”.

The paper is well written, it clearly states its theoretical guarantees and discusses intuition. In particular the privacy guarantees are clearly stated, and their difference to pure DP highlighted.  I thought Section 2.2 was especially well-written. I’m not an expert on sparse DP algorithms but it seemed to discuss prior work well, and highlight how this work is different, as well as why previous work did not immediately imply a solution.

It looks like the pooled mean does really well for false discovery rates, why is this?

==== Presentation

The paper is well written. I think the privacy aspect would be more compelling if the authors ran the propose test release version, which would actually be DP. It seems like this experiment would be interesting whether or not the propose test release version did well.

Small comments:
Definition 1 should be all measureable subsets, not just all subsets.
Typo: second sentence in intro: “large quantities of sensitive data are..” should be “large quantities of sensitive data have been..”
It would have been helpful to have the definition of the sgn of a sparse vecctor in the introduction, I was a little unsure exactly what was meant.
When defining the distribution space in (4), I think it would be helpful to state why the condition is required, not just that it is mild. It’s for Berry Esseen, yes?
It would be nice to see a discussion of the privacy implications of the five-fold cross-validation.
The second sentence of the abstract: it is not just common sense that randomness is required, unless the function is constant, randomness is provably required.

---

### Official Review · AnonReviewer1 · 2020-11-03
**Median DC for Sign Recovery: Privacy can be Achieved by Deterministic Algorithms**

**Rating:** 4
**Confidence:** 5

**Review:**

This paper considers the problem of private sign recovery for sparse mean estimation and sparse linear regression in a distributed setting. The paper proposes taking a coordinate-wise median among the reported local sign-vectors and gives its theoretical guarantees. Furthermore, the paper states that this is the first deterministic algorithm with a provable high-probability privacy guarantee.

I tend to vote for rejecting this paper for the following reasons:
1.  The problem considered in this paper is neither exciting nor complex. Furthermore, the algorithm proposed is quite natural. I agree these are some good results, but maybe not good enough for ICLR.

2. It is not appropriate to say this is the first deterministic algorithm with a provable high-probability privacy guarantee. In fact, utilizing robust estimators, as combined with propose-test-release (PTR) is a very basic technique in the literature of differential privacy. See Section 3.2 and 3.3 in the textbook (https://privacytools.seas.harvard.edu/files/privacytools/files/complexityprivacy_1.pdf). The intuition behind PTR is that for a robust estimator (like median), although the global sensitivity is huge, the local sensitivity is only large in some corner cases. And the local sensitivity is  negligible with high probability, where the randomness is drawn from the dataset's generation. Using robust estimators (which is deterministic), which gives privacy guarantees with high probability, is exactly the start point of these algorithms. Therefore, I do not find it appropriate to state it is a new observation. By the way, using the median is also a very classic idea in the robust estimation, for example, estimating high-dimensional Gaussians by utilizing Tukey median.

---

### Decision · Program_Chairs · 2021-01-07
**Final Decision**

**Decision:**

Reject

**Comment:**

The paper considers a problem of weak mean estimation under a differential privacy like constraint. Specifically, estimating the signs of a (sparse) mean, and not the actual values.

The reviewers brought up a number of concerns, including the weak privacy guarantee (a type of average-case privacy). Other lesser concerns include inaccuracies in comparisons with the literature and lack of interest in the algorithm/method itself.

As there was no response from the authors, there was little further discussion afterwards, and the reviewers remained in their opinion to reject the paper.